# Alterations in ACE and ACE2 Activities and Cardiomyocyte Signaling Underlie Improved Myocardial Function in a Rat Model of Repeated Remote Ischemic Conditioning

**DOI:** 10.3390/ijms222011064

**Published:** 2021-10-14

**Authors:** Beáta Bódi, Patrick M. Pilz, Lilla Mártha, Miriam Lang, Ouafa Hamza, Miklós Fagyas, Petra L. Szabó, Dietmar Abraham, Attila Tóth, Bruno K. Podesser, Attila Kiss, Zoltán Papp

**Affiliations:** 1Division of Clinical Physiology, Department of Cardiology, Faculty of Medicine, University of Debrecen, 4032 Debrecen, Hungary; bodibea0509@gmail.com (B.B.); lilla.martha@med.unideb.hu (L.M.); fagyasmiklos@med.unideb.hu (M.F.); atitoth@med.unideb.hu (A.T.); 2Center for Biomedical Research, Ludwig Boltzmann Institute for Cardiovascular Research, Medical University Vienna, 1090 Vienna, Austria; pmpilz@stanford.edu (P.M.P.); lang.miriam@gmx.net (M.L.); ouafa.hamza@meduniwien.ac.at (O.H.); petra.szabo@meduniwien.ac.at (P.L.S.); bruno.podesser@meduniwien.ac.at (B.K.P.); 3Center for Anatomy and Cell Biology, Medical University Vienna, 1090 Vienna, Austria; dietmar.abraham@meduniwien.ac.at; 4HAS-UD Vascular Biology and Myocardial Pathophysiology Research Group, Hungarian Academy of Sciences, 4032 Debrecen, Hungary

**Keywords:** repeated remote ischemic conditioning, cardiomyocyte mechanics, signaling

## Abstract

Post-ischemic left ventricular (LV) remodeling and its hypothetical prevention by repeated remote ischemic conditioning (rRIC) in male Sprague–Dawley rats were studied. Myocardial infarction (MI) was evoked by permanent ligation of the left anterior descending coronary artery (LAD), and myocardial characteristics were tested in the infarcted anterior and non-infarcted inferior LV regions four and/or six weeks later. rRIC was induced by three cycles of five-minute-long unilateral hind limb ischemia and five minutes of reperfusion on a daily basis for a period of two weeks starting four weeks after LAD occlusion. Sham operated animals served as controls. Echocardiographic examinations and invasive hemodynamic measurements revealed distinct changes in LV systolic function between four and six weeks after MI induction in the absence of rRIC (i.e., LV ejection fraction (LVEF) decreased from 52.8 ± 2.1% to 50 ± 1.6%, mean ± SEM, *p* < 0.05) and in the presence of rRIC (i.e., LVEF increased from 48.2 ± 4.8% to 55.2 ± 4.1%, *p* < 0.05). Angiotensin-converting enzyme (ACE) activity was about five times higher in the anterior LV wall at six weeks than that in sham animals. Angiotensin-converting enzyme 2 (ACE2) activity roughly doubled in post-ischemic LVs. These increases in ACE and ACE2 activities were effectively mitigated by rRIC. Ca^2+^-sensitivities of force production (pCa_50_) of LV permeabilized cardiomyocytes were increased at six weeks after MI induction together with hypophosphorylation of 1) cardiac troponin I (cTnI) in both LV regions, and 2) cardiac myosin-binding protein C (cMyBP-C) in the anterior wall. rRIC normalized pCa_50_, cTnI and cMyBP-C phosphorylations. Taken together, post-ischemic LV remodeling involves region-specific alterations in ACE and ACE2 activities together with changes in cardiomyocyte myofilament protein phosphorylation and function. rRIC has the potential to prevent these alterations and to improve LV performance following MI.

## 1. Introduction

Although rapid reperfusion has dramatically reduced short-term mortality of patients with acute myocardial infarction (MI), current management strategies cannot prevent the development of post-ischemic chronic heart failure in a significant proportion of MI cases [1]. Consequently, novel therapeutic strategies are sought to alleviate left ventricular (LV) remodeling and the long-term consequences of acute coronary syndrome.

Current pharmacological approaches aiming at post-ischemic heart failure (HF, including the administration of inhibitors of the renin–angiotensin–aldosterone system (RAAS) and of the *β*-adrenergic cascade) implicate roles for over-activated neuro-humoral signaling in myocardial remodeling [1]. During this maladaptive process, the myocardium develops morphological and functional alterations [2], whereby changes in the expression and phosphorylation of cardiomyocyte proteins ultimately render the heart limited in its mechanical performance. Of note, results of our earlier investigations helped to reveal mechanistic links between cardiomyocyte signaling and LV systolic and diastolic functions through the recognition of alterations in myofilament protein phosphorylation during myocardial hypertrophy and HF [3,4,5]. Hence, we hypothesize that myocardial protein alterations will also aid the elucidation of relevant signaling processes in the development of myocardial depression and during its prevention by novel experimental approaches following MI.

Remote ischemic conditioning (RIC), defined as a non-lethal ischemia/reperfusion (IR) insult in organs (e.g., liver, kidney) or body parts (e.g., limbs) distal to the heart confers protection against myocardial ischemia-reperfusion injury, as well as it reduces infarct size [6]. Up to now, numerous studies have demonstrated the cardioprotective potential and signaling of RIC against acute IR injury [7]; however, a recent large scale clinical trial failed to show the cardiac benefit of RIC in patients with ST-elevation MI [8]. Although an effect of RIC on post-infarct remodeling have been also implicated [9], the benefits of a single [10] or repeated RIC (rRIC) on long-term myocardial remodeling and the underlying mechanisms remain elusive [11]. Importantly, it has been shown that rRIC has the potential to improve coronary microcirculation in patients with congestive HF and to limit myocardial depression four weeks after MI in a rat model of experimental HF [12,13]. Nevertheless, coordination of the seemingly beneficial effects of rRIC remains largely unclear.

Here we focused on LV alterations in a rat model of post-ischemic remodeling and its prevention by rRIC. To these ends, tissue activities of the angiotensin-converting enzyme (ACE), and angiotensin-converting enzyme 2 (ACE2) were assayed. ACE is a well-known driver of adverse LV remodeling, hypertrophy and dilation following MI, and tissue angiotensin II level can critically depend on its local generation by myocardial ACE [14,15]. In contrast, ACE2 possesses anti-remodeling effects [16]. Hypothetically, the balance between tissue activities of ACE and ACE2 may parallel the commitment of the myocardium towards remodeling. To reveal critical signaling mechanisms (distal from ACE and ACE2) myofilament protein alterations and cardiomyocyte functions were also assessed here.

Our data illustrate rRIC with a potential to normalize myocardial ACE and ACE2 activities, cardiomyocyte signaling and function, and thereby to improve post-ischemic systolic performance.

## 2. Results

### 2.1. rRIC Improved LV Systolic Function

To induce myocardial ischemia, rat hearts were exposed via left thoracotomy and a ligature was placed around the LAD 2–3 mm away from its origin. Four weeks after MI induction, MI rats were left untreated (MI group) while in the MI + rRIC group a rRIC protocol was employed. rRIC was performed by 3 cycles of 5-min-long unilateral hind limb ischemia and 5 min of reperfusion once a day for 2 weeks. LV samples from sham operated animals (without LAD occlusion; Sham group) served as controls (Figure 1).

Histologic analyses of cardiac cross sections illustrated significant levels of collagen depositions in the anterior wall of LVs of infarcted animals consistent with the development of MI (6 weeks after LAD occlusion), and that rRIC did not influence infarct size (Figure 2A,B). Heart weight and heart weight to body weight ratios were markedly increased in rats with MI, and rRIC did not affect these parameters (Table 1). Systolic LV anterior wall thickness was similar at four and six weeks in the MI (2.1 ± 0.2 mm and 2.0 ± 0.2 mm, respectively) and MI + rRIC groups (2.2 ± 0.2 mm and 2.1 ± 0.1 mm, respectively) (Figure 2C). Changes in LVESD and LVEDD parameters were consistent with LV dilation following MI as revealed by transthoracic echocardiography (Figure 2D,E). Six weeks after MI induction, LVESD was significantly less (6.6 ± 0.5 mm) in the MI + rRIC group than in the MI group (7.9 ± 0.4 mm) suggestive for an improved systolic function upon rRIC (Figure 2D). LVEDD values were approximately the same in the MI and MI + rRIC groups at the end of the experimental protocol (Figure 2E).

Heart rates were similar in the three experimental groups (Sham: 232.5 ± 9.0 bpm; MI: 228.6 ± 4.5 bpm and MI + rRIC: 235.5 ± 8.2 bpm) in sedated animals at the end of the experiments (Figure 2F). Invasively measured LV peak systolic pressure (LVSP) and the maximal rate of rise of left ventricular pressure (+dP/dt_max_) were 109.8 ± 2.4 mmHg and 6791.3 ± 183.8 mmHg/s in the Sham group. The above parameters decreased in MI animals (i.e., to 86.2 ± 1.9 mmHg and 4621.8 ± 141.2 mmHg/s, respectively, *p* < 0.05), and also in the MI + rRIC group, although in the latter case only to a smaller degree (i.e., to 95.8 ± 2.9 mmHg and 5225.5 ± 96.3 mmHg/s, respectively, *p* < 0.05 vs. MI) at the same time point (Figure 2G,H). LV end-diastolic pressure (LVEDP) was 2.8 ± 0.3 mmHg in Sham animals, and it was higher in the MI (8.9 ± 0.9 mmHg, *p* < 0.05), and MI + rRIC (6.5 ± 0.8 mmHg) groups (Figure 2I).

VEF decreased from its baseline values (MI: 82.2 ± 1% and rRIC: 81.6 ± 0.7%) to MI: 52.8 ± 2.1% and rRIC: 48.2 ± 4.8% at four weeks after MI (Figure 3A,B). Following two additional weeks of daily RIC, LVEF increased to 55.2 ± 4.1% (*p* < 0.05) in the MI + rRIC group, while it declined to 50 ± 1.6% in the MI group (Figure 3B–D). LVEF was about the same in the Sham group at baseline, four weeks and six weeks (81.8 ± 0.6%; 79.8 ± 0.9%; 81.0 ± 0.6%) (Figure 3B). These morphological and functional data are consistent with an unaltered structure and diastolic function, but improved LV systolic function upon rRIC following MI.

### 2.2. ACE and ACE2 Activities in the Anterior and Inferior LV Areas

Mean tissue ACE activity in the LV anterior wall of the MI group (124.1 ± 15.2 pmol/min/mg) was about five times higher than that in the remote inferior wall (29.5 ± 2.6 pmol/min/mg, *p* < 0.05) or in the LV of the sham operated group (24.4 ± 1.3 pmol/min/mg, *p* < 0.05) (Figure 4A). MI related ACE upregulation of the LV anterior wall was markedly attenuated by rRIC (i.e., it was 72.0 ± 13.4 pmol/min/mg, *p* < 0.05 vs. MI anterior; *p* < 0.05 vs. MI inferior). ACE activity in the inferior wall (18.6 ± 19.7 pmol/min/mg) was apparently not affected by rRIC (Figure 4B).

Mean ACE2 activity was significantly higher in both the anterior and the remote inferior LV areas in the MI group (11 ± 0.6 pmol/min/mg and 9.4 ± 1 pmol/min/mg, respectively) than that in the Sham group (5.5 ± 0.24 pmol/min/mg, *p* < 0.05) (Figure 4C,D). rRIC significantly attenuated ACE2 activities in the inferior (5.6 ± 0.38 pmol/min/mg, *p* < 0.05 vs. MI), but not in the anterior LV area of the MI + rRIC group (7.9 ± 1.2 pmol/min/mg, *p* > 0.05 vs. MI) (Figure 4C,D).

### 2.3. Differences in Ca^2+^-Regulated Cardiomyocyte Force Production upon rRIC

Ca^2+^-activated force production (F_active_) in LV permeabilized cardiomyocytes was determined at different Ca^2+^ concentrations in vitro. Means of maximal Ca^2+^-activated force (F_max_, pCa = 4.75) values were similar in the three experimental groups for the infarcted anterior (Sham: 14.69 ± 0.71 kN/m^2^; MI: 13.63 ± 1.60 kN/m^2^; MI + rRIC: 13.51 ± 0.80 kN/m^2^) and remote inferior areas (Sham: 18.27 ± 1.62 kN/m^2^; MI: 15.32 ± 1.35 kN/m^2^; MI + rRIC: 14.75 ± 1.13 kN/m^2^) (Figure 5A,B).

Nevertheless, a leftward shift in the normalized pCa–force relationships indicated increased Ca^2+^-sensitivities of force production for the cardiomyocytes in both investigated LV regions following MI (Figure 5C,D). Accordingly, means of pCa_50_ values were significantly higher in the anterior and inferior LV walls in the MI group (pCa_50_: 6.05 ± 0.03 and 6.11 ± 0.03, respectively) than in Sham animals (pCa_50_: 5.94 ± 0.01 and 5.93 ± 0.04, respectively). rRIC treatment prevented the development of the increase in pCa_50_ in the anterior and inferior regions in the MI + rRIC group (5.93 ± 0.01 and 6.00 ± 0.02, respectively) (Figure 5C–F).

### 2.4. rRIC Influenced Myofilament Protein Phosphorylation after Ischemic Injury

Site-specific phosphorylation studies on myofilament proteins were performed to elucidate the molecular background of the higher pCa_50_ values in LV cardiomyocytes. Phosphorylation levels of cTnI at the PKA-specific Ser-22/23 and PKC-specific Thr-144 sites were significantly lower in the infarcted anterior LV region (0.79 ± 0.05 and 0.58 ± 0.06, respectively, in relative units) than those in the Sham group (1.00 ± 0.03 and 1.00 ± 0.09, respectively, in relative units) (Figure 6A,C). Similar to the infarcted zone, a reduced phosphorylation level at Ser-22/23 site was recognized in the inferior LV wall (Sham: 1.01 ± 0.04 vs. MI: 0.82 ± 0.03, in relative units) (Figure 6B). Nevertheless, the extent of phosphorylation at the Thr-144 site in the inferior LV wall did not differ significantly among the three experimental groups (Sham: 1.00 ± 0.04; MI: 1.09 ± 0.06; MI + rRIC: 1.03 ± 0.04, in relative units) (Figure 6D). rRIC prevented the hypophosphorylation of cTnI at the Ser-22/23 site in the anterior (1.09 ± 0.04, in relative units) and inferior regions (1.08 ± 0.08, in relative units) and at the Thr-144 site in the anterior LV wall (1.00 ± 0.08, in relative units, *p* < 0.05) (Figure 6A–C).

Overall phosphorylation level in cMyBP-C was significantly lower in cardiomyocytes of the anterior LV region in the MI group (0.66 ± 0.05, in relative units, Figure 7A) than those in the same area of the Sham group (1.00 ± 0.04, in relative units, Figure 7A). Following rRIC the level of cMyBP-C phosphorylation (0.97 ± 0.07, in relative units) could not be distinguished from that in the Sham group (Figure 6A). No significant differences in the total phosphorylation levels of cMyBP-C of the inferior LV region in the three experimental groups (Sham: 1.00 ± 0.03; MI: 1.18 ± 0.05; MI + rRIC: 1.13 ± 0.11, in relative units) could be recognized (Figure 7B).

### 2.5. Titin-Dependent Stiffness of LV Cardiomyocytes Was Not Changed after Myocardial Infarction

Ca^2+^-independent passive tension (F_passive_) of cardiomyocytes was not affected either in the anterior or in the inferior areas in the MI group (1.39 ± 0.21 kN/m^2^ and 1.07 ± 0.18 kN/m^2^, respectively) (Figure 8A,B). These values were largely comparable to those in the Sham group (1.25 ± 0.24 kN/m^2^ and 0.98 ± 0.17 kN/m^2^, respectively) and to those in the MI + rRIC group (1.34 ± 0.21 kN/m^2^ and 0.88 ± 0.09 kN/m^2^, respectively) (Figure 8A,B).

Total phosphorylation levels of titin were also similar in the three different experimental groups of cardiomyocytes and in the two investigated regions (Sham: 1.01 ± 0.06; MI: 0.95 ± 0.07; MI + rRIC: 1.09 ± 0.05 in the LV anterior region; Sham: 1.03 ± 0.16; MI: 0.92 ± 0.03; MI + rRIC: 1.05 ± 0.07, in the LV inferior region, all in relative units) (Figure 8C,D).

## 3. Discussion

Our results present new evidence for post-ischemic myocardial remodeling and its prevention by rRIC in rat hearts. Here we show that LV performance is tightly coupled to a set of interacting variables: tissue activities of ACE and ACE2, the Ca^2+^-sensitivity of force production, and myofilament protein phosphorylation six weeks after MI. Our data highlight links between systemic cardiovascular and local cardiomyocyte signaling that can be exploited during the management of post-ischemic cardiomyopathy. To our best knowledge, no previous studies employing rRIC protocols have addressed the above variables [17,18].

Systolic and diastolic functions of the infarcted LV chamber depend on the extents of myocardial necrosis, fibrosis, ventricular dilation, and the integrity of the remaining myocardium. An increase in ventricular wall stress, the level of neuro-humoral stimulation (e.g., by the RAAS, β-adrenergic system) together with local tissue factors and other mechanisms contribute to adverse LV remodeling and eventually to a progressive decline in LV performance following MI [14,19]. Here, we paid specific attention to the remaining myocardium, its function and signaling, but not to infract size or fibrosis. To these ends, myocardial parameters in two different LV regions of the post-ischemic heart were screened: the anterior wall (i.e., in the vicinity of the MI), and the inferior wall (i.e., in a remote location from the MI) with the aim to reveal hypothetical signaling inhomogeneities within the post-ischemic LV. Moreover, our protocol of rRIC aimed at the prevention of LV remodeling, rather than limiting necrosis, since infarct size is not the sole determinant of long-term clinical outcomes following MI [20].

In contrast to acute models of RIC [6,11], infarct size was not affected by rRIC in our chronic model, because scar formation was probably complete by the time rRIC was initiated in this study (i.e., at four weeks after MI induction). However, two weeks later, changes in LVEF and LVESD were consistent with a modest improvement of the systolic LV function by rRIC that was not seen in its absence. This observation motivated us for the investigation of the signaling processes mobilized by rRIC. Interestingly, rRIC largely reduced tissue ACE and ACE2 activities in comparison to those in the MI group, and hence our data support direct links between the dysregulation of tissue ACE and ACE2 activities and myocardial remodeling as has been also illustrated by previous results of our group and others [21,22,23,24,25]. The mechanism by which rRIC lead to the downregulation of cardiac ACE and ACE2 activities was not aimed in this study. Nevertheless, based on the apparent close relationship between ACE and ACE2 activities and their expressions in lung and heart tissues, a reduction in the expression levels of these enzymes cannot be excluded upon rRIC [26,27].

Cardiomyocyte systolic and diastolic functions depend on the interplay between cytosolic [Ca^2+^] and contractile myofilaments. Here we found that Ca^2+^-sensitivity of force production (pCa_50_) was significantly increased in permeabilized cardiomyocytes in both the anterior and in the inferior regions of the post-ischemic LV chamber. An increase in pCa_50_ is considered as a hallmark of myocardial remodeling and HF in humans and its animal models [3,4,28]. pCa_50_ is the function of cTnI phosphorylation and cMyBP-C phosphorylation. In this context, protein kinase A (PKA) phosphorylates Ser-22/23 (Ser-23/24 in humans) residues of cTnI and multiple phosphorylation sites of cMyBP-C, and thereby decreases pCa_50_ upon *β*-adrenergic stimulation [28,29,30]. Here we screened phosphorylation levels in both above myofilament proteins and found that cTnI phosphorylation at the Ser-22/23 sites was significantly reduced in the anterior and posterior LV walls of infarcted animals, suggestive for a systemic effect for the entire LV. A reduced level of cMyBP-C phosphorylation could also be recognized in the anterior wall of infarcted LVs but not in the inferior regions. Collectively, our findings are in line with downgraded *β*-adrenergic cardiomyocyte signaling at six weeks after LAD ligature that lead to the hypophosphorylations of cTnI and (in the vicinity of MI) of cMyBP-C, and consequently to an increase in Ca^2+^-sensitivity of force production.

Here we also paid attention to the Thr-144 site of cTnI which is thought to be phosphorylated specifically by protein kinase C (PKC) upon the activations of type 1 angiotensin II receptors (and of the RAAS). Phosphorylation of cTnI at Thr-144 is expected to increase pCa_50_ [31]. Interestingly, we observed a reduction in Thr-144 phosphorylation in the anterior LV wall, but not in the inferior wall following MI. This observation, together with our ACE and Ca^2+^-sensitivity data, imply that the relatively large elevation in cardiac tissue ACE activity was associated with a downgraded cardiomyocyte responsiveness for angiotensin II too, and that the PKA specific Ser-22/23 cTnI phosphorylation sites acted as primary regulators of pCa_50_ in the post-ischemic rat heart. Of note, all changes in pCa_50_, as well as in protein phosphorylations were effectively antagonized by rRIC. In line with our findings, a previous study also implicated the benefit of RIC on myocardial contractile function [32].

Taken together, the number of alterations in the anterior wall outweighed that of the inferior wall in post-ischemic rat hearts. These findings implicate that besides systemic neuro-humoral signaling (e.g., by the RAAS and *β*-adrenergic system) local regulators also affect cardiomyocyte remodeling in the vicinity of myocardial infarction. Moreover, our data also imply that systolic cardiomyocyte function declines with an increase in pCa_50_ along with the development of relative insensitivities for the β-adrenergic system and the RAAS. Conversely, the moderate improvement in systolic LV function upon rRIC was paralleled by normalizations in ACE and ACE2 activities, pCa_50_ values, PKA and PKC specific protein phosphorylations, and hence these data implicated regained myocardial tissue sensitivities for the *β*-adrenergic system and also for the RAAS.

F_max_, F_passive_, and phosphorylation of sarcomeric titin were not affected either by MI or by rRIC in this study. Accordingly, these variables could not be involved in the phenomena investigated here.

Certain limitations of our study need to be acknowledged. The rRIC protocol was not optimized for maximal hemodynamic effects. Hence, larger responses upon different conditioning methods cannot be excluded Our approach relied on preclinical and clinical evidence that revealed pivotal roles for local ACE and ACE2 activities in the formation and elimination of angiotensin II during post-ischemic cardiac remodeling [33,34]. However, we did not attempt to reconstruct the local RAAS in the heart, and therefore, the involvement of additional tissue factors affecting local angiotensin peptide levels and/or their receptors cannot be excluded. Moreover, we did not focus on intracellular Ca^2+^ cycling of cardiomyocytes, although β-adrenergic and RAAS dependent alterations in the regulation of the intracellular Ca^2+^ transient (e.g., by phospholamban phosphorylation) are also involved during rRIC [35]. Finally, extrapolation of our data to human hearts can be complicated not only by species differences but also by the health conditions (i.e., co-morbidities and co-medications) of patients suffering in is-chemic heart disease [36].

In summary, post-ischemic LV remodeling involves the massive upregulation of tissue ACE activity in the vicinity of MI and, to a smaller degree, ACE2 in the entire post-ischemic LV chamber. Results of our site-specific phosphorylation assays pointed to probable reductions in cardiomyocyte *β*-adrenergic and RAAS responsiveness. rRIC prevented these changes and subsequently improved LV function. Results of the CONDI-2/ERIC-PPCI study revealed that infarct size reduction is difficult to achieve by a single acute RIC protocol in the clinical setting [8]. While our delayed and repeated RIC method could not limit infarct size either, it did improve cardiomyocyte function and signaling. Hence we postulate that preservation of the integrity of cardiomyocyte β-adrenergic and RAAS signaling can serve to prevent post-ischemic LV remodeling in the remaining myocardium and can potentially improve clinical outcomes. Accordingly, rRIC protocols should be optimized in terms of timing, cycle numbers and duration. Clearly, further preclinical and clinical investigations are needed to verify the translational value of the implicated links between rRIC, myocardial signaling and clinical outcomes.

## 4. Materials and Methods

### 4.1. Experimental Specimens and Ethical Statement

Male Sprague–Dawley (SD) rats (12-week-old at the beginning of the experiments) were housed in standard cages at 22 ± 2 °C controlled room temperatures with natural day and night exchange. Standard laboratory chow and tap water were available ad libitum. The experimental protocol was approved by the regional Ethics Committee for Laboratory Animal Experiments at the Medical University of Vienna and the Austrian Ministry of Science Research and Economy (BMWFW-66.009/0023-WF/V/3b/2016). All procedures conform to the guidelines from ARRIVE and Directive 2010/63/EU of the European Parliament on the protection of animals used for scientific purposes.

### 4.2. Experimental Protocol for rRIC

Myocardial infarction was induced by the permanent ligation of the left anterior descending artery (LAD) as described previously [9]. Briefly, rats were anaesthetized by intraperitoneal injection of a mixture of Xylazine (4 mg/kg; Bayer, Germany) and Ketamine (100 mg/kg; Dr. E. Gräub AG, Switzerland), intubated (14-gauge tube) and ventilated (0.9 mL/kg body weight, 75–85 stroke/min). Rectal temperature was measured and maintained at 37.5–38.5°C by a heated operating table. Myocardial infarction was associated with pallor of the myocardial area at risk and ST-elevation on ECG signal. Analgesia was initiated by intraperitoneal injection of Piritramide (0.1 mL/kg body weight) preoperatively and Piritramide in drinking water was applied as a postoperative analgesic regimen (2 ampules of Piritramide with 30 mL of Glucose 5% in 250 mL water).

Functional and biochemical measurements in LV cardiac tissues and cardiomyocytes were performed at the age of 18 weeks in vivo and ex vivo. One day after echocardiographic assessments LV hemodynamic function was invasively monitored as described previously [37]. For this purpose, rats were anaesthetized by intraperitoneal injection of a mixture of Xylazin (4 mg/kg; Bayer, Germany) and Ketamin (100 mg/kg; Dr E. Gräub AG, Switzerland) and intubated. Following hemodynamic determinations, the chest was opened, and the heart was extracted. The heart was cut on the mid-papillary muscle level and the base was used for histology and the apex was used for biochemical assessment. Small blocks of tissue samples were cut from the infarcted anterior and non-infarcted inferior (remote) area and dissected in cold isolating solution (ISO) (1 mM MgCl2, 100 mM KCl, 2 mM EGTA, 4 mM ATP, 10 mM imidazole; pH 7.0, 0.5 mM phenylmethylsulfonyl fluoride, 40 μM leupeptin and 10 μM E-64, all from Sigma-Aldrich, St. Louis, MO, USA) then snap frozen in liquid nitrogen.

### 4.3. Transthoracic Echocardiography

Transthoracic echocardiography was performed as described previously [38]. Briefly, rats were anesthetized (isoflurane 2–3%) and echocardiography was performed using Vivid7 system (GE Healthcare, USA) equipped with an 11.5 MHz 10S sector transducer. Left ventricular ejection fraction (LVEF), left ventricular end-diastolic and end-systolic diameter (LVEDD and LVESD, respectively) were evaluated at a midpapillary short axis view. The range of post-ischemic hemodynamic parameters (e.g., LVEF) was similar to that found by others [39].

### 4.4. Histological Analyses

Formalin-fixed paraffin-embedded tissue sections were hematoxylin and eosin (HE) stained. The extent of fibrosis in cardiac muscle sections was visualized by Masson Goldner staining (Masson-Goldner staining kit, Sigma-Aldrich/Merck, Darmstadt, Germany) as described previously [18]. Images were acquired by microscopy (Olympus VS120 Virtual Slide Microscope System; Olympus, Tokyo, Japan) and captured by digital camera (AVT PIKE F-505C VC 50; Allied Vision Technologies, Stadtroda, Germany). To evaluate scar formation at 6 weeks after MI, the LV area from the mid part of the myocardium at mid-papillary muscle level was used. The percentage of fibrosis was acquired with Adobe Photoshop Element (Adobe Photoshop, Version 14.1) based on the equation: %fibrosis (scar formation) = fibrotic area/(fibrotic area + non-fibrotic area of LV).

### 4.5. Assessment of ACE and ACE2 Activities

ACE and ACE2 activities in cardiac (infarcted and non-infarcted) tissue samples were measured as described previously [40,41]. Briefly, tissue samples were weighed, and a proportional amount of 100 mM tris(hydroxymethyl)aminomethane hydrochloride (TRIS) buffer (pH 7.0) was added then homogenized. The tissue homogenates were centrifuged at 13,000 rpm for 5 min and the protein concentration of the supernatant was determined by PierceTM BCA Protein Assay Kit (Thermo Scientific, Waltham, MA, USA) using TECAN (SparkControl Magellan V2.2) plate reader. ACE activity was determined with an artificial substrate (Abz-FRK(Dnp)P-OH (Enzo Life Science, Exeter, UK) in a reaction mixture (total volume of 210 µL) containing 6 µL of 1 mg/mL tissue homogenates (35-fold dilution) in 100 mM TRIS buffer, 50 mM NaCl, 10 µM ZnCl_2_. ACE2 activity was assessed with a fluorogenic substrate, Mca-APK(Dnp) (Enzo Life Science, Exeter, UK). The reaction mixture (total volume of 210 µL) contained 6 µL of 1 mg/mL tissue homogenates (35-fold dilution) in reaction buffer. The reaction buffer involved a protease inhibitor cocktail 10 μM Bestatin-hidrochloride (Sigma-Aldrich, St. Louis, MO, USA), 10 μM Z-prolyl-prolinal (Enzo Life Science, Exeter, UK), 5 μM Amastatin-hydrochloride (Sigma-Aldrich, St. Louis, MO, USA), 10 μM Captopril (Sigma-Aldrich, St. Louis, MO, USA), 5 mM NaCl, 100 μM ZnCl2, 75 mM TRIS HCl, pH 6.5.

Measurements were performed in 96-well plates (Greiner-Bio One) at 37 °C. The fluorescence intensity change was detected by TECAN (SparkControl Magellan V2.2) plate reader, λ_ex_ was 340 nm and λ_em_ was 405 nm in case of the ACE activity measurement and λ_ex_ was 320 nm and λ_em_ was 405 nm in case of the ACE2 activity assessment. The changes in fluorescence intensity were detected in kinetic loops, at 1-min intervals for at least 30 min and the intensity values were plotted as a function of reaction time. The fluorescence intensity values were fitted by a linear regression (GraphPad Software, San Diego, CA, USA), and the fit with the data was accepted only when r^2^ was >0.9. ACE and ACE2 activities were calculated by the following equation: activity = (S/k) * D/P; where S is the rate of the increase in fluorescence intensity (1/min), k is the change of fluorescence intensity during the complete cleavage of 1 pmol Abz-FRK(Dnp)P-OH or Mca-APK(Dnp) substrate, respectively, D is the dilution of the sample and P is the concentration of protein expressed in mg/mL. One unit (U) means 1 pmol substrate cleavage in 1 min by 1 mg of protein.

### 4.6. Contractile Force Measurements in Permeabilized Cardiomyocytes

LV tissue samples from anterior and non-infarcted remote inferior areas were mechanically disrupted in ISO and membrane-permeabilized by 0.5% Triton X-100 detergent in ISO at 4°C. Myocyte-sized preparations were mounted with silicone adhesive (DAP 100% all-purpose silicone sealant; Baltimore, MD, USA) to two stainless steel insect needles, which were connected to a sensitive force transducer (SensoNor, Horten, Norway) and to an electromagnetic high-speed length controller (Aurora Scientific Inc., Aurora, Canada) in ISO at 15°C. Subsequent cardiomyocyte isometric force generation was recorded at sarcomere length of 2.3 μm and analyzed by LabVIEW software (National Instruments, Corp., Austin, TX, USA). Cardiomyocyte Ca^2+^-activated force generation was induced by transferring the preparation from a relaxing (containing 37.11 mM KCl, 10 mM BES, 6.41 mM MgCl_2_, 7 mM EGTA, 6.94 mM Na_2_ATP, 15 mM Na_2_CrP, 40 µM leupeptin, 10 µM E64; pH 7.2) to an activating solution (same content as relaxing solution, supplemented with Ca^2+^-EGTA). The Ca^2+^ concentrations ([Ca^2+^]) expressed in pCa units were estimated by –lg [Ca^2+^]. The pCa of the activating and relaxing solutions was 4.75 and 9.0, respectively. When a steady force level had been reached, a rapid release–restretch maneuver (30 ms) was applied to determine the baseline of the force generation and hence the Ca^2+^-activated total force (F_total_). Ca^2+^- independent passive force (F_passive_) of the cardiomyocytes was measured by shortening to 80% of the original preparation length at pCa 9.0 for 8 s. The active force (F_active_) was calculated as a difference of the F_total_ and Fpassive. Maximal activation at pCa 4.75 was used to determine the maximal Ca^2+^-activated isometric force (F_max_), while activations with intermediate [Ca^2+^] (pCa 5.4– 7.0) yielded the pCa–isometric force relationship. Isometric forces at submaximal [Ca^2+^] normalized to Fmax and fitted to a modified Hill-equation (Origin 6.0, Microcal Software, Northampton, MA. USA) and to determine the Ca^2+^-sensitivity of force production (pCa_50_). Original forces of every individual cell were normalized to myocyte cross sectional-area, calculated by the width and height of the cardiomyocyte. Absolute force values of F_max_, F_active_, and F_passive_ were expressed in kN/m^2^. LV samples from four different hearts (*n* = 8 cardiomyocytes from each heart) were used in the force measurements.

### 4.7. Biochemical Analysis of Myofilament Proteins

Cardiomyocytes were isolated and permeabilized from frozen LV tissue samples and were homogenized in sample buffer (8 M urea, 2 M thiourea, 3% (*w*/*v*) sodium dodecyl sulfate (SDS), 75 mM DTT, 50 mM Tris-HCl, pH 6.8, 10% (*v*/*v*) glycerol, bromophenol blue, 40 µM leupeptin and 10 µM E-64) for 45 min by vortex procedure. After centrifugation (16,000× *g* for 5 min at 24°C), protein amount of supernatant was estimated by dot–blot technique and set to 2 mg/mL concentration correlating the sample to bovine serum albumin (BSA) standards. Agarose- strengthened 2% and 4% SDS-polyacrilamide gels were used to separate N2B titin (~3200 kDa) and cardiac myosin-binding protein C (cMyBP-C). Overall phosphorylation levels of titin and cMyBP-C proteins were assessed by Pro-Q^®^ Diamond phosphoprotein staining (Invitrogen, Molecular Probes, Eugene, OR, USA) according to the manufacturer’s protocol, while total protein levels were assayed by Coomassie blue (Reanal, Budapest, Hungary).

For site-specific cardiac troponin I (cTnI) phosphorylation Western immunoblot assays, proteins were separated in 12% polyacrilamide gels, then transferred to nitrocellulose membrane and probed with PKA- [Ser-22/23 (1:1000)], or PKC-dependent primary [Thr-144 (1:500), Abcam, Cambridge, UK] and secondary antibodies [peroxidase-conjugated secondary antibody (1:300), anti-rabbit IgG from Sigma-Aldrich, St. Louis, MO, USA]. Proteins were detected by enhanced chemiluminescence and documented by MF-ChemiBIS 3.2 gel documentation system (DNR Bio-Imaging Systems, Ltd., Jerusalem, Israel). All biochemical analyses were performed on 4–6 samples/groups.

### 4.8. Statistical Analysis

Data are expressed as mean ± SEM. Statistical analysis was performed with GraphPad Prism 6.0 software (GraphPad Software Inc., San Diego, CA, USA). Western immunoblot assays were performed in triplicate. Signal intensities of protein bands were quantified by using the ImageJ (National Institutes of Health, Bethesda, Maryland, USA) and Magic Plot (Magicplot Systems, Saint Petersburg, Russia) software. Differences were evaluated by one-way ANOVA followed by Bonferroni post hoc test or Students’ paired *t*-test when appropriate. *p* values of <0.05 were considered statistically significant.

## Figures and Tables

**Figure 1 ijms-22-11064-f001:**
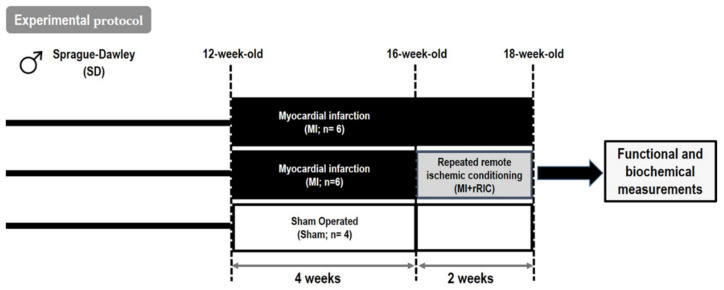
Scheme of the experimental protocol. The left anterior descending coronary artery (LAD) was ligated to induce myocardial infarction (MI group, *n* = 6) in 12-week-old male rats. rRIC was performed by 3 cycles of 5-min-long unilateral hind limb ischemia and 5 min of reperfusion once a day for 2 weeks, 4 weeks after LAD ligation in a separate group of animals (MI + rRIC group; *n* = 6). Sham operated animals (Sham group, *n* = 4) served as controls. Myocardial tissues were collected at 18 weeks for functional and biochemical measurements.

**Figure 2 ijms-22-11064-f002:**
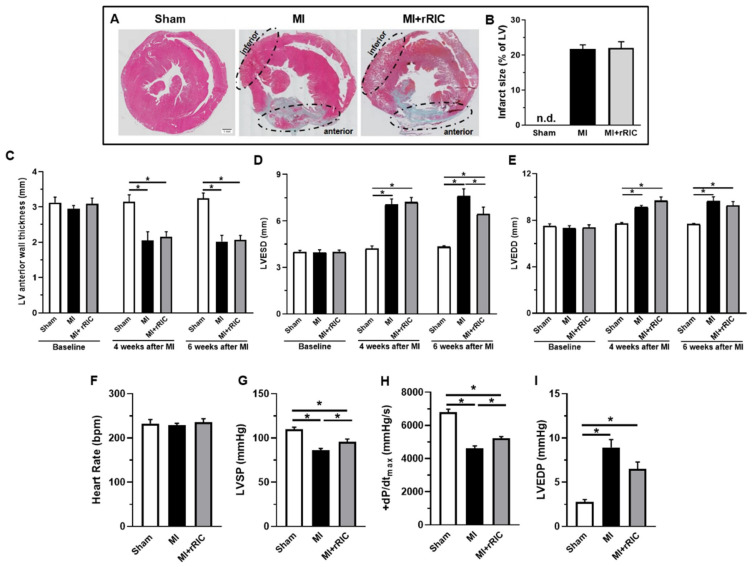
Histological analysis and functional characteristics of the infarcted hearts. Representative Masson’s trichrome-stained cross-sections are shown from the Sham, MI, and MI + rRIC groups (**A**). Red staining identifies viable myocardial tissues without scar formation in the inferior region (non-infarcted area) of the left ventricle (LV), while blue staining shows scar formation in the anterior region (infarcted area) of LV following MI. Quantitative aspects of myocardial infarct sizes are given in a bar graph (**B**). Time courses of systolic LV anterior wall thickness and LV dimensions at baseline, 4 or 6 weeks after MI induction as determined by echocardiography. Panels (**C**–**E)** illustrate systolic LV anterior wall thickness, LV end-systolic diameter (LVESD), and LV end-diastolic diameter (LVEDD) in the Sham, MI and MI + rRIC groups. Heart rate, LV peak systolic pressure (LVSP), maximal rate of rise of LV pressure ( + dP/dt_max_), and LV end-diastolic pressure (LVEDP) at six weeks after MI induction are also given for the three experimental groups panels (**F**–**I**). Values are given as mean ± SEM; *n* = 4–6 animals/group. * *p* < 0.05.

**Figure 3 ijms-22-11064-f003:**
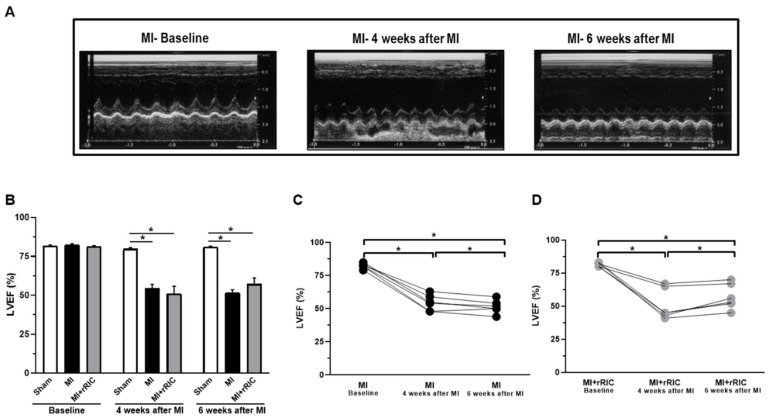
Changes in LV ejection fraction (LVEF) upon rRIC. Representative echocardiographic images during the determinations of LVEF (**A**). LVEF at baseline, 4 weeks and 6 weeks after MI induction (**B**). LVEF of individual rats in the MI group (**C**) and in the MI + rRIC group (**D**) at baseline, 4 and 6 weeks after MI induction. Values are given as mean ± SEM; *n* = 4–6 animals/group. * *p* < 0.05.

**Figure 4 ijms-22-11064-f004:**
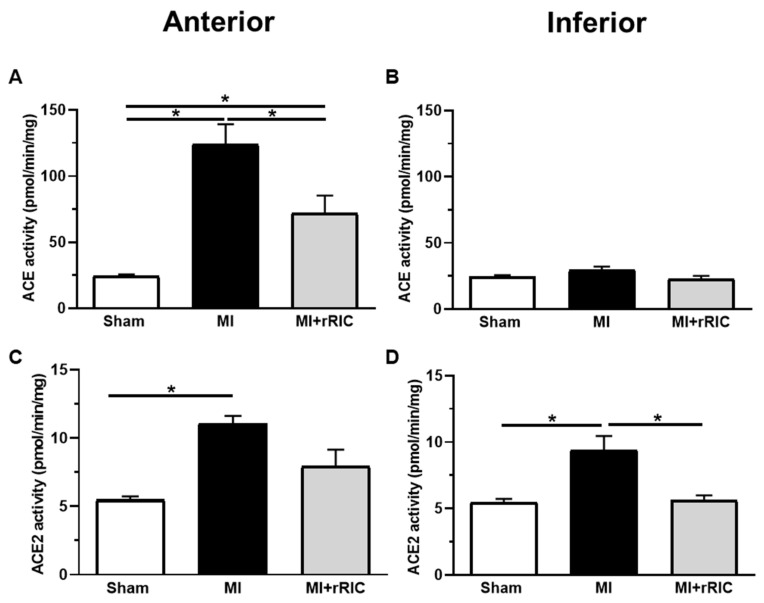
Myocardial ACE and ACE2 activities in cardiac tissue samples 6 weeks after MI induction. ACE (**A**,**B**) and ACE2 activities (**C**,**D**) were determined in anterior (**A**,**C**) and inferior (**B**,**D**) cardiac tissue samples from all experimental groups. Values are given as mean ± SEM; *n* = 4–6 rats/group. * *p* < 0.05.

**Figure 5 ijms-22-11064-f005:**
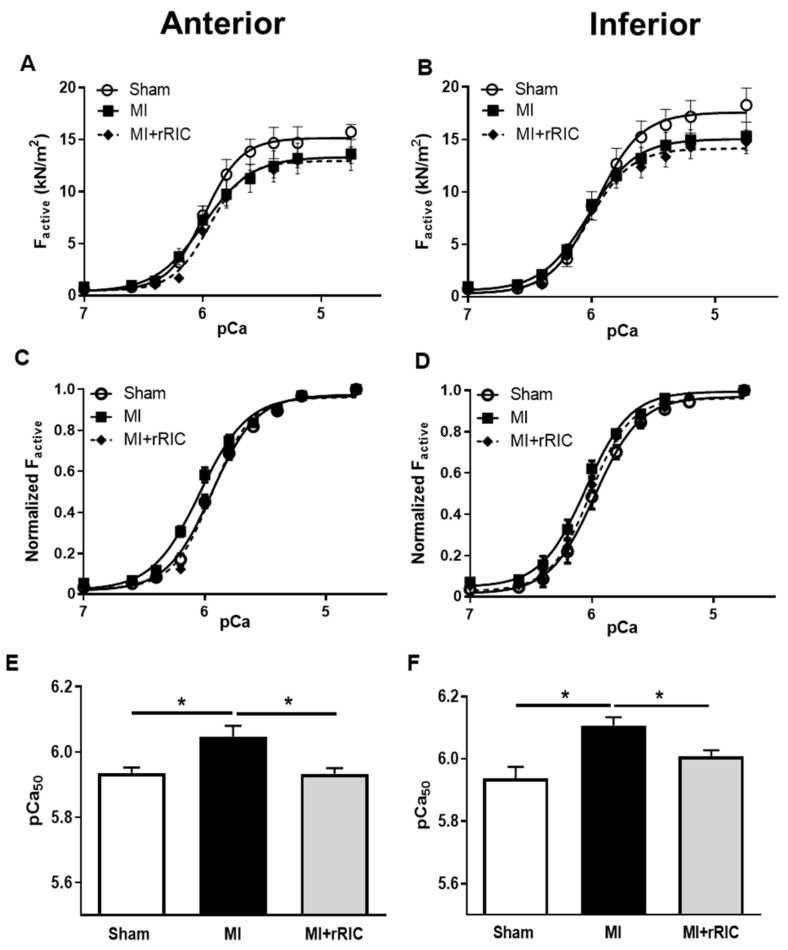
Ca^2+^-sensitivity of force production (pCa_50_) in isolated, permeabilized cardiomyocytes of the anterior (**A**,**C**,**E**) and inferior LV areas (**B**,**D**,**F**). Absolute values of activate force (F_active_) production in isolated myocyte-sized preparations at different Ca^2+^ concentrations in the anterior (**A**) and inferior (**B**) areas from the Sham, MI, and MI + rRIC groups. Sarcomere length was adjusted to 2.3 µm. F_active_ values at submaximal Ca^2+^ concentrations (pCa 5.4–7.0) were expressed relative to F_max_ (pCa 4.75) to determine normalized pCa-force relationships (**C**,**D**). Bar graphs (**E**,**F**) highlight the midpoints of normalized pCa–force relationships. Data are expressed as mean ± SEM, * *p* < 0.05; *n* = 8 from at least four different hearts.

**Figure 6 ijms-22-11064-f006:**
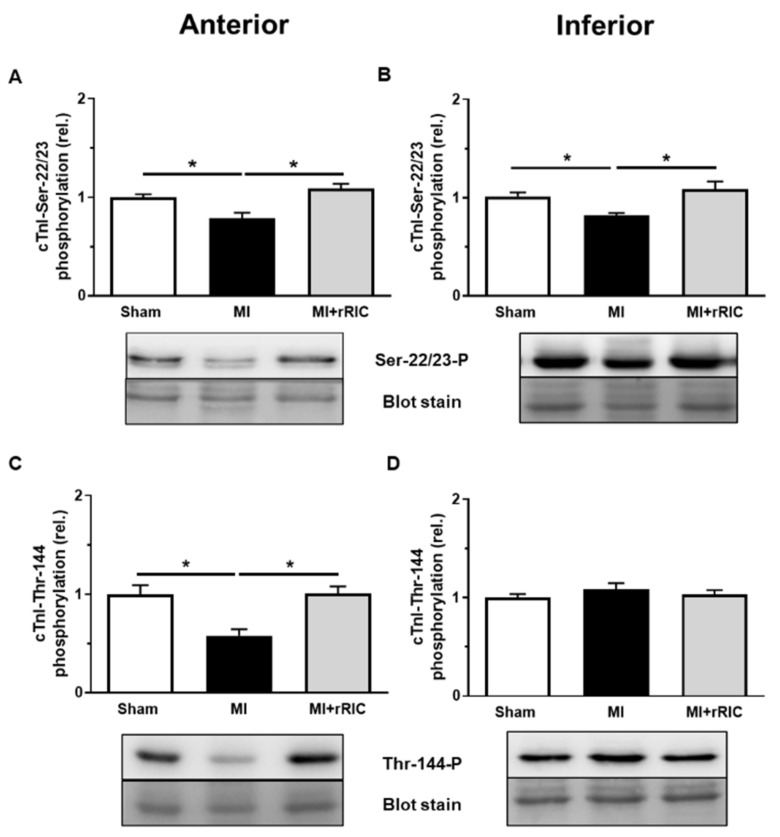
PKA- and PKC-specific phosphorylation of cTnI in the anterior (**A**,**C**) and inferior (**B**,**D**) LV regions. Cardiac troponin-I (cTnI) phosphorylation levels at the PKA-specific serine-22/23 (Ser-22/23-P) (**A**,**B**) and PKC-specific threonine-144 (Thr-144-P) (**C**,**D**) residues were screened by Western immunoblotting in Sham, MI, and MI + rRIC groups of anterior and inferior LV areas. The phosphorylation sites of cTnI were labelled with specific antibodies. Total protein amounts were assessed by a super sensitive blot stain. Bars represent means ± SEM (normalized to the mean of the Sham group as control), * *p* < 0.05, *n* = 7–10 independent determinations from at least 4–5 different hearts.

**Figure 7 ijms-22-11064-f007:**
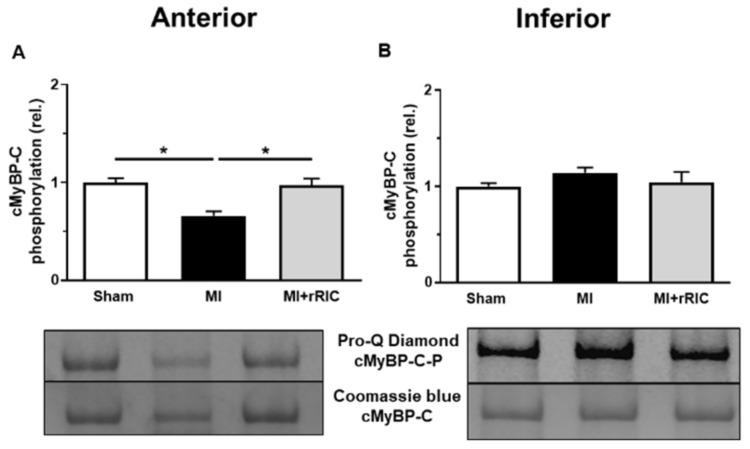
cMyBP-C phosphorylation in cardiomyocytes in the anterior (**A**) and inferior (**B**) LV regions. Total phosphorylation levels of cMyBP-C were investigated with Pro-Q^®^ Diamond phosphoprotein staining. The upper bands illustrate the phosphorylation status of cMyBP-C and the lower bands indicate the total protein amounts for each representative membrane (*n* = 5–13 independent determinations from at least 4–5 different heart samples). Values are given as mean ± SEM, * *p* < 0.05 vs. control (Sham) groups.

**Figure 8 ijms-22-11064-f008:**
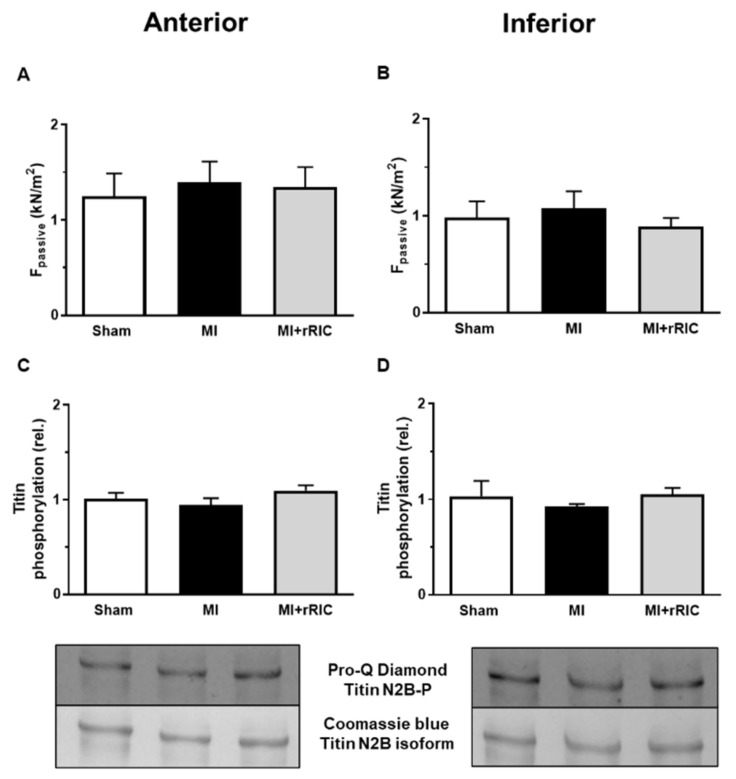
Titin-based passive tension and titin protein phosphorylation in the anterior (**A**,**C**) and inferior (**B**,**D**) LV regions. Titin-dependent passive tension (F_passive_) in permeabilized cardiomyocytes was determined at a sarcomere length of 2.3 μm (**A**,**B**) (*n* = 8–10 cardiomyocytes/groups). Total phosphorylation status of the N2B titin isoform was assessed by Pro-Q^®^ Diamond phosphoprotein staining. Total titin amount was visualized by Coomassie-blue staining (**C**,**D**). Protein homogenates were collected from four control and six infarcted hearts and assays were repeated four to six times. Data are given as mean ± SEM.

**Table 1 ijms-22-11064-t001:** Characteristics of the experimental groups.

Parameter/Group	Sham	MI	MI + rRIC
*n*	4	6	6
BW (g)	448 ± 13	448 ± 5	439 ± 5
HW (g)	1.37 ± 0.02	1.78 ± 0.06 *	1.62 ± 0.02 *
HW/BW (g)	2.97 ± 0.10	3.80 ± 0.15 *	3.70 ± 0.05 *
infarct size (%)	n.a.	21.72 ± 1.17 *	22.02 ± 1.78 *

BW: body weight; HW: heart weight; HW/BW: heart to body weight ratio on the day of sacrifice (i.e., 6 weeks after myocardial infarction). Values are given as mean ± SEM; * *p* < 0.01 vs. the Sham group, n.a.: not applicable.

## Data Availability

The data presented in this study are available on request from the first or corresponding author.

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
