# Peer review of "Alterations in ACE and ACE2 Activities and Cardiomyocyte Signaling Underlie Improved Myocardial Function in a Rat Model of Repeated Remote Ischemic Conditioning"

_ijms, 2021, doi:10.3390/ijms222011064_

Round 1
Reviewer 1 Report
Authors investigated the efficiency of remote ischemic conditioning for male Sprague-Dawley rat model of myocardial infarction. Authors showed interesting mechanism of beta-adrenergic cardiomyocyte signaling including Ca2+ sensitivity. However, some results lack of significance, theory, and reliability. The reviewer listed concerns. Please consider and reply.
- Please show the representative figures of transthoracic echocardiogram at systolic and diastolic phases to confirm whether the measurement is correct or not because authors measured by only short axis. In addition, authors should show anterior wall thickness in Figure 2 and baseline LVEF in Figure 2F-G. The range of LVEF was so broad that reviewer thought the quality of surgery was not stable. Authors should consider making exclusion criteria of LVEF post MI to resolve this concern.
- According to results, the context and significance of ACE/ACE2 were unclear. In addition, the association between ACE/ACE2 and remote ischemic conditioning were not examined. What did authors intend to measure with ACE/ACE2? Generally, Angiotensin II (1-8) primarily plays key role on fibrosis. Authors must show the expression of angiotensin II and angiotensin 1-9 if authors investigate ACE1 and ACE2, respectively. It is no meanings to investigate only ACE/ACE2 or ATII receptor.
- Overall, remote ischemic conditioning seemed to prevent the reduction in cardiomyocyte beta-adrenergic responsiveness. In contrast, there were no pathological changes in cardiac tissue. Consequently, the reviewer thought that measured hearts just hypercontracted due to beta-adrenergic responsiveness. Then, this study lost the significance and incorrect the conclusion. Please check another view of echocardiogram including long axis at least or measure the peak value of LV diastolic pressure.
- Authors should show all significant changes in Figure 3.
Author Response
Reviewer #1:
Authors investigated the efficiency of remote ischemic conditioning for male Sprague-Dawley rat model of myocardial infarction. Authors showed interesting mechanism of beta-adrenergic cardiomyocyte signaling including Ca2+ sensitivity. However, some results lack of significance, theory, and reliability. The reviewer listed concerns. Please consider and reply.
Response: Thank you for considering part of our work interesting. We are also grateful for your critical remarks because they largely improved our manuscript.
1/1 Please show the representative figures of transthoracic echocardiogram at systolic and diastolic phases to confirm whether the measurement is correct or not because authors measured by only short axis.
Response: Thank you for this comment. We accepted your request, and now we show representative echocardiograms at systolic and diastolic phases for the MI group (at baseline, 4 weeks and 6 weeks) to confirm the correctness of our measurements (new Figure 3A). Additionally, now we also included new hemodynamic data from our model study (i.e. heart rate, LV peak systolic pressure (LVSP), the maximal rate of rise of left ventricular pressure (+dP/dtmax) and LV end-diastolic pressure (LVEDP)) in Figure 2F-2I. Collectively, these data show that rRIC had no effects on infarct size, heart size, heart rate, but it improved systolic function (LVESD, LVSP, +dP/dtmax; (Figure 2) and EF (Figure 3)).
Change in text (Abstract, Results, Materials and Methods, Figures): “Echocardiographic examinations and invasive hemodynamic measurements revealed distinct changes in LV systolic function between four and six weeks after MI induction in the absence of rRIC (e.g. LV ejection fraction (LVEF) decreased from 52.8±2.1% to 50±1.6%, mean±SEM, P<0.05) and in the presence of rRIC (LVEF increased from 48.2±4.8% to 55.2±4.1%, P<0.05).” (Abstract)
“Heart rates were similar in the three experimental groups (Sham: 232.5±9.0 bpm; MI: 228.6±4.5 bpm and MI+rRIC: 235.5±8.2 bpm) in sedated animals at the end of the experiments (Figure 2F). Invasively measured LV peak systolic pressure (LVSP) and the maximal rate of rise of left ventricular pressure (+dP/dtmax) were 109.8±2.4 mmHg and 6791.3±183.8 mmHg/s in the Sham group. The above parameters decreased in MI animals (i.e. to 86.2±1.9 mmHg and 4621.8±141.2 mmHg/s, respectively, P<0.05), and also in the MI+rRIC group, although in the latter case only to a smaller degree (i.e. to 95.8±2.9 mmHg and 5225.5±96.3 mmHg/s, respectively) at the same time point (Figure 2G and H). LV end-diastolic pressure (LVEDP) was 2.8±0.3 mmHg in Sham animals, and it was higher in the MI (8.9±0.9 mmHg, P<0.05), and MI+rRIC (6.5±0.8 mmHg) groups (Figure 2I).” (Results)
“Functional and biochemical measurements in LV cardiac tissues and cardiomyocytes were performed at the age of 18 weeks in vivo and ex vivo. One day after the echocardiographic assessments LV hemodynamic function was invasively monitored as described previously [37]. For this purpose, rats were anaesthetized by intraperitoneal injection of a mixture of Xylazin (4 mg/kg; Bayer, Germany) and Ketamin (100 mg/kg; Dr E. Gräub AG, Switzerland) and intubated. Following hemodynamic determinations, the chest was opened, and the heart was extracted.” (Materials and Methods)
New Figure 2F-2I.
New reference [37]: Osmanagic-Myers, S.; Kiss, A.; Manakanatas, C.; Hamza, O.; Sedlmayer, F.; Szabo, P. L.; Fischer, I.; Fichtinger, P.; Podesser, B. K.; Eriksson, M.; Foisner, R., Endothelial progerin expression causes cardiovascular pathology through an impaired mechanoresponse. J Clin Invest 2019, 129, (2), 531-545.
1/2 In addition, authors should show anterior wall thickness in Figure 2 and baseline LVEF in Figure 2F-G.
Response: We accepted your request, and now we also show anterior wall thickness in Figure 2C and included baseline LVEF values in Figure 3C-3D (formerly Figure 2F-G).
Change in text (Results, Figures): “Systolic LV anterior wall thickness was similar at four and six weeks in the MI (2.1±0.2 mm and 2.0±0.2 mm, respectively) and MI+rRIC groups (2.2±0.2 mm and 2.1±0.1 mm, respectively) (Figure 2C). Changes in LVESD and LVEDD parameters were consistent with LV dilation following MI as revealed by transthoracic echocardiography (Figure 2D and E).”
“LVEF decreased from its baseline values (MI: 82.2±1% and rRIC: 81.6±0.7%) to MI: 52.8±2.1% and rRIC: 48.2±4.8% at four weeks after MI (Figure 3B). Following two additional weeks of daily RIC, LVEF increased to 55.2±4.1% (P<0.05) in the MI+rRIC group, while it declined to 50±1.6% in the MI group (Figure 3B, 3C and 3D). LVEF was about the same in the Sham group at baseline, four weeks and six weeks (81.8±0.6%; 79.8±0.9%; 81.0±0.6%) (Figure 3B). These morphological and functional data are consistent with an unaltered structure and diastolic function, but improved LV systolic function upon rRIC following MI.”
Extended Figure 2 and new Figure 3.
1/3 The range of LVEF was so broad that reviewer thought the quality of surgery was not stable. Authors should consider making exclusion criteria of LVEF post MI to resolve this concern.
Response: We understand your concern, but respectfully disagree on the instability of surgery quality. The scatter in hemodynamic parameters typically increases following the induction of MI in animal models of this kind, and the range of postischemic parameters found in our study was comparable to that found by others (e.g. see Table 2 in Ghotbi et al. J Nucl Cardiol. 2019 Jun;26(3):798-809). Therefore, we did not consider making exclusion criteria of LVEF post MI.
Change in text (Materials and Methods): “The range of postischemic hemodynamic parameters (e.g. LVEF) was similar to that found by others [39].”
New reference [39]: Ghotbi, A. A.; Clemmensen, A.; Kyhl, K.; Follin, B.; Hasbak, P.; Engstrøm, T.; Ripa, R. S.; Kjaer, A., Rubidium-82 PET imaging is feasible in a rat myocardial infarction model. J Nucl Cardiol 2019, 26, (3), 798-809.
2/1 According to results, the context and significance of ACE/ACE2 were unclear. In addition, the association between ACE/ACE2 and remote ischemic conditioning were not examined. What did authors intend to measure with ACE/ACE2?
Response: We fully agree and accept this criticism. By reading your note we realized that ACE/ACE2 ratio does not add much (on top of ACE and ACE2 activities) to this paper. Therefore, we decided to leave ACE/ACE2 ratios out from Figure 4 (formerly Figure 3) and from the manuscript.
Change in text (Figures, Legends): Figure 4C (formerly Figure 3C) has been omitted.
2/2 Generally, Angiotensin II (1-8) primarily plays key role on fibrosis. Authors must show the expression of angiotensin II and angiotensin 1-9 if authors investigate ACE1 and ACE2, respectively. It is no meanings to investigate only ACE/ACE2 or ATII receptor.
Response: We understand your concern and anticipate the significance of local angiotensin II and other angiotensin peptide levels. However, based on preclinical and clinical evidence for ACE inhibitor therapy in postischemic cardiac remodelling (e.g. Paul et al. Physiol Rev 2006, 86, (3), 747-803. Hirsch et al., Circ Res 1991, 69, (2), 475-82) we opted for the determination of tissular ACE and ACE2 activities only. Here we hypothesized that the modulation of local RAAS is involved in the cardioprotection evoked by rRIC and aimed at recognizing alterations in myocardial signaling and function. Nevertheless, we did not attempt to reconstruct the local RAAS in the heart. These aspects are now emphasised in the limitation section of this paper.
Change in text (Discussion, limitation section): “Our approach relied on preclinical and clinical evidence that revealed pivotal roles for local ACE and ACE2 activities in the formation and elimination of angiotensin II during postischemic cardiac remodeling [33, 34]. However, we did not attempt to reconstruct the local RAAS in the heart, and therefore, the involvement of additional tissue factors affecting local angiotensin peptide levels and/or their receptors cannot be excluded.”
New references [33]: Paul, M.; Poyan Mehr, A.; Kreutz, R., Physiology of local renin-angiotensin systems. Physiol Rev 2006, 86, (3), 747-803.
[34] Hirsch, A. T.; Talsness, C. E.; Schunkert, H.; Paul, M.; Dzau, V. J., Tissue-specific activation of cardiac angiotensin converting enzyme in experimental heart failure. Circ Res 1991, 69, (2), 475-82.
3/1 Overall, remote ischemic conditioning seemed to prevent the reduction in cardiomyocyte beta-adrenergic responsiveness. In contrast, there were no pathological changes in cardiac tissue. Consequently, the reviewer thought that measured hearts just hypercontracted due to beta-adrenergic responsiveness. Then, this study lost the significance and incorrect the conclusion. Please check another view of echocardiogram including long axis at least or measure the peak value of LV diastolic pressure.
Response: We understand your concern and therefore now we included heart rate (HR), LV peak systolic pressure (LVSP), maximal rate of rise of left ventricular pressure (+dP/dtmax) and LV end-diastolic pressure (LVDP) (Figure 2). The unchanged HR and non-significant decrease in LVDP together with increased LVSP and +dP/dtmax upon rRIC are in line with the results of echocardiography (i.e. LVEF) and with an improved LV systolic function following rRIC, but argue against the possibility of hypercontraction.
Change in text (Results): “Heart rates were similar in the three experimental groups (Sham: 232.5±9.0 bpm; MI: 228.6±4.5 bpm and MI+rRIC: 235.5±8.2 bpm) in sedated animals at the end of the experiments (Figure 2F). Invasively measured LV peak systolic pressure (LVSP) and the maximal rate of rise of left ventricular pressure (+dP/dtmax) were 109.8±2.4 mmHg and 6791.3±183.8 mmHg/s in the Sham group. The above parameters decreased in MI animals (i.e. to 86.2±1.9 mmHg and 4621.8±141.2 mmHg/s, respectively, P<0.05), and also in the MI+rRIC group, although in the latter case only to a smaller degree (i.e. to 95.8±2.9 mmHg and 5225.5±96.3 mmHg/s, respectively) at the same time point (Figure 2G and H). LV end-diastolic pressure (LVEDP) was 2.8±0.3 mmHg in Sham animals, and it was higher in the MI (8.9±0.9 mmHg, P<0.05), and MI+rRIC (6.5±0.8 mmHg) groups (Figure 2I).”
New experimental data are included in Figure 2.
4/ Authors should show all significant changes in Figure 3.
Response: We understand and accept your request and now we included comparisons between all groups in the Results section describing Figure 4 (former Figure 3). Moreover, by reading your request we also recognized that the way how ACE and ACE2 data are presented should be similar to that used in Figures 5-8. Therefore, we rearranged the layout of Figure 4 (formerly Figure 3).
Change in text (Results): “Mean tissue ACE activity in the LV anterior wall of the MI group (124.1±15.2 pmol/min/mg) was about five times higher than that in the remote inferior wall (29.5±2.6 pmol/min/mg, P<0.05) or in the LV of the sham operated group (24.4±1.3 pmol/min/mg, P<0.05) (Figure 4A). MI related ACE upregulation of the LV anterior wall was markedly attenuated by rRIC (i.e. it was 72.0±13.4 pmol/min/mg, P<0.05 vs. MI anterior; P<0.05 vs. MI inferior). ACE activity in the inferior wall (18.6±19.7 pmol/min/mg) was apparently not affected by rRIC (Figure 4B).
Mean ACE2 activity was significantly higher in both the anterior and the remote inferior LV areas in the MI group (11±0.6 pmol/min/mg and 9.4±1 pmol/min/mg, respectively) than that in the Sham group (5.5±0.24 pmol/min/mg, P<0.05, Figure 4C). rRIC significantly attenuated ACE2 activities in the inferior (5.6±0.38 pmol/min/mg, P<0.05 vs. MI), but not in the anterior LV area of the MI+rRIC group (7.9±1.2 pmol/min/mg, P>0.05 vs. MI, Figure 4D).”
New Figure 4.
Reviewer 2 Report
Alterations in ACE and ACE2 activities and cardiomyocyte signaling underlie improved myocardial function in a rat model of repeated remote ischemic conditioning
Bódi, Pilz et al. investigated the left ventricular remodeling process following myocardial infarction in a rat model. Also, they have studied the effects of repeated remote ischemic conditioning, to alleviate the detrimental cardiac effects. Importantly, the chronic animal model has high clinical relevance, and they have used scientifically sound techniques to investigate the remodeling process.
Questions:
- While the methods used during the research are all of high value individually, the connection between the findings and the overall conclusion should be emphasized more clearly. Providing a summary figure or graphical abstract would be useful and would help to communicate the main findings of the study.
- Besides ACE and ACE2 enzyme activity, measuring the protein levels would be also interesting. Please provide Western blot experiments for this.
- Phosphorilation of cTnI and cMyBP-C were measured and discussed in detail. In this regard, phosphorylation of phospholamban would be also interesting to be measured experimentally.
- The authors have used repeated remote ischemic conditioning to alleviate the effects of chronic myocardial infarction and subsequent cardiac remodeling. Please emphasize in the “Discussion” section the clinical relevance of rRIC and the translational value. Also, please discuss the fact that rRIC was unable to decrease infarct size in this experimental setting. Please discuss findings from the literature about the differences between RIC in acute and chronic models, and also the discrepancies between RIC in the preclinical studies (where most of the studies show infarct size reduction) and in the clinical setting, where RIC failed to confer significant benefits in the CONDI-2/ERIC-PPCI study (which is referenced in the “Introduction” section).
Author Response
Reviewer #2:
Bódi, Pilz et al. investigated the left ventricular remodeling process following myocardial infarction in a rat model. Also, they have studied the effects of repeated remote ischemic conditioning, to alleviate the detrimental cardiac effects. Importantly, the chronic animal model has high clinical relevance, and they have used scientifically sound techniques to investigate the remodeling process.
Response: Thank you for considering our work relevant. We are also grateful for your critical remarks because they helped us to improve our manuscript.
2/1 While the methods used during the research are all of high value individually, the connection between the findings and the overall conclusion should be emphasized more clearly. Providing a summary figure or graphical abstract would be useful and would help to communicate the main findings of the study.
Response: Thank you for this request. We have prepared a graphical abstract for the illustration of the main findings.
Change in text: We have included a graphical abstract.
2/2 Besides ACE and ACE2 enzyme activity, measuring the protein levels would be also interesting. Please provide Western blot experiments for this.
Response: We agree that ACE and ACE2 protein levels are of relevance. Nevertheless, in two of our recent investigations (where the above relationships were studied in lung and myocardial tissues in detail) we found a close relationship between the expression levels of these enzymes and their activities. Therefore, we did not include similar characterizations here.
Source: Figure 4B in Bánhegyi et al., Cells. 2021 Jul 6;10(7):1708. doi: 10.3390/cells10071708. https://www.mdpi.com/2073-4409/10/7/1708/htm
Soruce: Fagyas et al., Geroscience, in press)
Change in text (Discussion): “Nevertheless, based on the apparent close relationship between ACE and ACE2 activities and their expressions in lung and heart tissues, a reduction in the expression levels of these enzymes cannot be excluded upon rRIC [26, 27].”
New references [26]: Bánhegyi, V.; Enyedi, A.; Fülöp, G.; Oláh, A.; Siket, I. M.; Váradi, C.; Bottyán, K.; Lódi, M.; Csongrádi, A.; Umar, A. J.; Fagyas, M.; Czuriga, D.; Édes, I.; Pólos, M.; Merkely, B.; Csanádi, Z.; Papp, Z.; Szabó, G.; Radovits, T.; Takács, I.; Tóth, A., Human Tissue Angiotensin Converting Enzyme (ACE) Activity Is Regulated by Genetic Polymorphisms, Posttranslational Modifications, Endogenous Inhibitors and Secretion in the Serum, Lungs and Heart. Cells 2021, 10, (7).
[27] Fagyas, M.; Bánhegyi, V.; Úri, K.; Enyedi, A.; Lizanecz, E.; Mányiné Siket, I.; Mártha, L.; Fülöp, G. Á.; Radovits, T.; Pólos, M.; Merkely, B.; Kovács, Á.; Szilvássy, Z.; Ungvári, Z.; Édes, I.; Csanádi, Z.; Boczán, J.; Takács, I.; Szabó, G.; Balla, J.; Balla, G.; Seferović, P. M.; Papp, Z.; Tóth, A., Changes in the SARS-CoV-2 cellular receptor ACE2 levels in cardiovascular patients: a potential biomarker for the stratification of COVID-19 patients. GeroScience 2021, Accepted by Publisher.
2/3 Phosphorilation of cTnI and cMyBP-C were measured and discussed in detail. In this regard, phosphorylation of phospholamban would be also interesting to be measured experimentally.
Response: We agree. Nevertheless, analyses of intracellular Ca2+ cycling was beyond the scope of this study. This is now acknowledged in the limitation section of the Discussion.
Change in text (Discussion): “Moreover, we did not focus on intracellular Ca2+ cycling of cardiomyocytes, although β-adrenergic and RAAS dependent alterations in the regulation of the intracellular Ca2+ transient (e.g. by phospholamban phosphorylation) are also involved during rRIC [35].”
New reference: [35] Inserte, J.; Hernando, V.; Ruiz-Meana, M.; Poncelas-Nozal, M.; Fernández, C.; Agulló, L.; Sartorio, C.; Vilardosa, U.; Garcia-Dorado, D., Delayed phospholamban phosphorylation in post-conditioned heart favours Ca2+ normalization and contributes to protection. Cardiovasc Res 2014, 103, (4), 542-53.
2/4 " The authors have used repeated remote ischemic conditioning to alleviate the effects of chronic myocardial infarction and subsequent cardiac remodeling. Please emphasize in the “Discussion” section the clinical relevance of rRIC and the translational value. Also, please discuss the fact that rRIC was unable to decrease infarct size in this experimental setting. Please discuss findings from the literature about the differences between RIC in acute and chronic models, and also the discrepancies between RIC in the preclinical studies (where most of the studies show infarct size reduction) and in the clinical setting, where RIC failed to confer significant benefits in the CONDI-2/ERIC-PPCI study (which is referenced in the “Introduction” section).
Response: We are very grateful for this comment, because it stimulated us to put our results more into context of previous research efforts employing RIC.
Change in text (Discussion): “Here, we paid specific attention to the remaining myocardium, its function and signaling, but not to infract size or fibrosis.”
“Moreover, our protocol of rRIC aimed at the prevention of LV remodeling, rather than limiting necrosis, since infarct size is not the sole determinant of long-term clinical outcomes following MI [20].”
“In contrast to acute models of RIC [6, 11], infarct size was not affected by rRIC in our chronic model, because scar formation was probably complete by the time rRIC was initiated in this study (i.e. at four weeks after MI induction). However, two weeks later, changes in LVEF and LVESD were consistent with a modest improvement of the systolic LV function by rRIC that was not seen in its absence.”
“Results of the CONDI-2/ERIC-PPCI study revealed that infarct size reduction is difficult to achieve by a single acute RIC protocol in the clinical setting [8]. While our delayed and repeated RIC method could not limit infarct size either, it did improve cardiomyocyte function and signaling. Hence we postulate, that preservation of the integrity of cardiomyocyte β-adrenergic and RAAS signaling can serve to prevent post-ischemic LV remodeling in the remaining myocardium and can potentially improve clinical outcomes. Accordingly, rRIC protocols should be optimized in terms of timing, cycle numbers and duration. Clearly, further preclinical and clinical investigations are needed to verify the translational value of the implicated links between rRIC, myocardial signaling and clinical outcomes.”
New reference [20]: Traverse, J. H.; Swingen, C. M.; Henry, T. D.; Fox, J.; Wang, Y. L.; Chavez, I. J.; Lips, D. L.; Lesser, J. R.; Pedersen, W. R.; Burke, N. M.; Pai, A.; Lindberg, J. L.; Garberich, R. F., NHLBI-Sponsored Randomized Trial of Postconditioning During Primary Percutaneous Coronary Intervention for ST-Elevation Myocardial Infarction. Circ Res 2019, 124, (5), 769-778.

Round 2
Reviewer 1 Report
Authors resolved the reviewer's concerns. But please pay attention to RAAS because ACE and ACE2 cannot affect without substrates.
Reviewer 2 Report
All issues have been addressed.